# Real-world data from a molecular tumor board demonstrates improved outcomes with a precision N-of-One strategy

Shumei Kato ⓘ et al.#

Next-generation sequencing (NGS) can identify novel cancer targets. However, interpreting the molecular findings and accessing drugs/clinical trials is challenging. Furthermore, many tumors show resistance to monotherapies. To implement a precision strategy, we initiated a multidisciplinary (basic/translational/clinical investigators, bioinformaticians, geneticists, and physicians from multiple specialties) molecular tumor board (MTB), which included a project manager to facilitate obtaining clinical-grade biomarkers (blood/tissue NGS, specific immunohistochemistry/RNA expression including for immune-biomarkers, per physician discretion) and medication-acquisition specialists/clinical trial coordinators/navigators to assist with medication access. The MTB comprehensively reviewed patient characteristics to develop N-of-One treatments implemented by the treating physician's direction under the auspices of a master protocol. Overall, 265/429 therapy-evaluable patients (62%) were matched to ≥1 recommended drug. Eighty-six patients (20%) matched to all drugs recommended by MTB, including combinatorial approaches, while 38% received physician's choice regimen, generally with unmatched approach/low degree of matching. Our results show that patients who receive MTB-recommended regimens (versus physician choice) have significantly longer progression-free (PFS) and overall survival (OS), and are better matched to therapy. High (≥50%) versus low (<50%) Matching Score therapy (roughly reflecting therapy matched to ≥50% versus <50% of alterations) independently correlates with longer PFS (hazard ratio [HR], 0.63; 95% confidence interval [CI], 0.50–0.80; $P < 0.001$) and OS (HR, 0.67; 95% CI, 0.50–0.90; $P = 0.007$) and higher stable disease ≥6 months/partial/ complete remission rate (52.1% versus 30.4% $P < 0.001$) (all multivariate). In conclusion, patients who receive MTB-based therapy are better matched to their genomic alterations, and the degree of matching is an independent predictor of improved oncologic outcomes including survival.

---

#A list of authors and their affiliations appears at the end of the paper.

Next-generation sequencing (NGS) has allowed the identification of novel potential targets for patients with cancer. Examples of successful matching of tumor molecular alterations to cognate drugs include: NTRK inhibitors larotrectinib and entrectinib in multiple solid tumors with *NTRK* fusions[1,2], ROS1 inhibitors entrectinib and crizotinib in non-small cell lung cancer (NSCLC) with *ROS1* alterations[3,4], and the FGFR inhibitor erdafitinib in *FGFR*-altered urothelial carcinoma[5]. Unfortunately, the majority of patients still do not benefit from single-agent targeted therapies, and most patients who do respond eventually develop resistance[6–8].

Biologic factors that may limit responsiveness to matched targeted monotherapies include genomic heterogeneity and complexity, as well as the fact that advanced tumors often have unique (N-of-One) molecular profiles[9]. Moreover, it may be challenging to differentiate driver from passenger molecular alterations in tumors with complicated genomic portfolios[10–14]. Several lines of evidence suggest that optimized therapy may require a customized/personalized combinatorial approach[15,16].

In order to facilitate implementation of a precision medicine strategy in the clinic, we initiated a multidisciplinary molecular tumor board (MTB)[17–19]. The MTB represents a vehicle to integrate a comprehensive review of the patient characteristics, including clinical history, imaging, pathology, laboratory results, and molecular profiling, in the presence of specialists from diverse medical fields with the expertise of basic and translational scientists and computational biologists. The goal of the MTB was to develop an N-of-One treatment plan that could be initiated by the patient's physician under the auspices of a master protocol, with the assistance of clinical trial coordinators/navigators and medication acquisition specialists to facilitate drug availability. Of note, the MTB served as an advisory board, with the final decision made by the patient's physician. Hence, our MTB reflects a real-world experience with molecular profiling and patient treatment in the context of an academic medical center. Herein, we present the outcome of 715 patients with advanced cancer presented at our MTB, and demonstrate that adherence to MTB recommendations in order to match patients with targeted therapies was associated with higher degrees of matching and improved outcomes.

## Results

**Patient characteristics**. Among 715 patients with diverse malignancies, the median age was 61 years (range: 3–92 years), and 58.7% (N = 420) were women. Patients had advanced/metastatic disease, and the majority of patients had ≥2 prior therapies. The most common diagnosis was breast cancer (18% [129/715]), followed by colorectal cancer (12.2% [87/715]), hematologic malignancies (7.1% [51/715]), gastroesophageal (7.1% [51/715]) and pancreatic cancer (6.7% [48/715]) (Table 1). The physician, per their clinical judgment of necessity, ordered molecular tests. Physicians often presented patients upon receipt of molecular results, regardless of whether or not progressive disease was apparent. In general, the physician did not change therapy unless there was progressive disease on the current therapy or the current therapy was not tolerable.

Overall, 429 patients were evaluable for therapy after MTB discussion. Patients were not evaluable mostly because they received no further therapy after the MTB discussion or because their therapy was not changed within six months after the MTB (see Supplementary Fig. 1).

**Utilization of variety of molecular profiling laboratories.** During the MTB discussion, all profiling reports were included for the discussion if the testing was performed at a CLIA-certified

### Table 1 Baseline demographics of patients presented at the Molecular Tumor Board (N = 715 unique patients)[a].

| | |
|---|---|
| Period | December 2012–September 2018 |
| Number of meetings | 200 |
| Age | Median, 61 years; Range, 3–92 years |
| Gender, N (%) | Men, 295 (41.3%); Women, 420 (58.7%) |
| Number of physicians who presented ≥1 case | 58 |
| **Diagnosis, N (%)** | |
| Breast cancer | 129 (18.0) |
| Colorectal cancer | 87 (12.2) |
| Hematologic malignancies | 51 (7.1) |
| Gastroesophageal cancer | 51 (7.1) |
| Pancreatic cancer | 48 (6.7) |
| Biliary cancer | 37 (5.2) |
| Lung cancer | 36 (5.0) |
| Gynecologic cancer | 36 (5.0) |
| Other GI malignancies[b] | 33 (4.6) |
| Sarcoma | 32 (4.5) |
| Bladder/Ureter cancer | 20 (2.8) |
| Head and neck cancer | 19 (2.7) |
| Hepatocellular carcinoma | 18 (2.5) |
| Neuroendocrine malignancies | 17 (2.4) |
| Prostate cancer | 13 (1.8) |
| CNS malignancies | 12 (1.7) |
| Thyroid cancer | 5 (0.7) |
| Other malignancies | 71 (9.9) |

[a]Among the 858 presented patients, 99 patients were presented more than once. Only the first records were included in the analysis. Forty-four patients without complete discussion record were excluded (Supplementary Fig. 1). Only patients in face-to-face Molecular Tumor Board meetings were included.
[b]Other GI malignancies include appendiceal adenocarcinoma (N = 27), duodenal cancer (N = 4) and small bowel adenocarcinoma (N = 2).
CNS central nervous system, GI gastrointestinal.

(clinical-grade) laboratory. Tissue NGS was performed on 646 patients at seven different laboratories. Blood-derived cell-free DNA (cfDNA) was evaluated in 309 patients at two laboratories. Additionally, mRNA expression analysis, immunohistochemistry (IHC) and immunotherapy-associated markers (tumor mutation burden [TMB], microsatellite instability [MSI], PD-L1) were evaluated in selected cases (N = 39, N = 115, N = 362 respectively) (Supplementary Table 1).

Through tissue NGS (N = 646), *TP53* was the most commonly altered (52.3% [338/646]) followed by *KRAS* (23.8% [154/646]), *PIK3CA* (15.8% [102/646]), *APC* (13.6% [88/646]) and *CDKN2A/B* (11.5% [74/646]) (Fig. 1a and Supplementary Table 2). Among patients who had cfDNA analysis (N = 309), the most common alterations were seen in *TP53* (48.9% [151/309]) followed by *KRAS* (22.3% [69/309]), *PIK3CA* (18.4% [57/309]), *BRAF* (12.3% [38/309]) and *EGFR* (12.0% [37/309]) (Fig. 1b and Supplementary Table 3).

Among patients who had selected IHC analysis (N = 115), RRM1 was negative in 83% (33/40), TOP2A was positive in 78% (29/37), and ERCC1 was negative in 71% (37/52) of patients (see Supplementary Table 4 for interpretation of results and Supplementary Fig. 2). Thirty-nine patients had selected mRNA profiling and showed various expression patterns (e.g., TOP2A high in 83.3% [10/12], ERCC1 low in 73.3% [11/15], TS low in 61.5% [8/13]) (Supplementary Table 5).

Among patients tested for targets associated with immunotherapeutic implications (N = 362), TMB was high in 5.7% (16/280) (definition of TMB-high varied from each laboratory, e.g., Foundation One defined it as ≥20 mutations/megabase while Caris defined it as ≥17 mutations/megabase). MSI was high in 3.2%

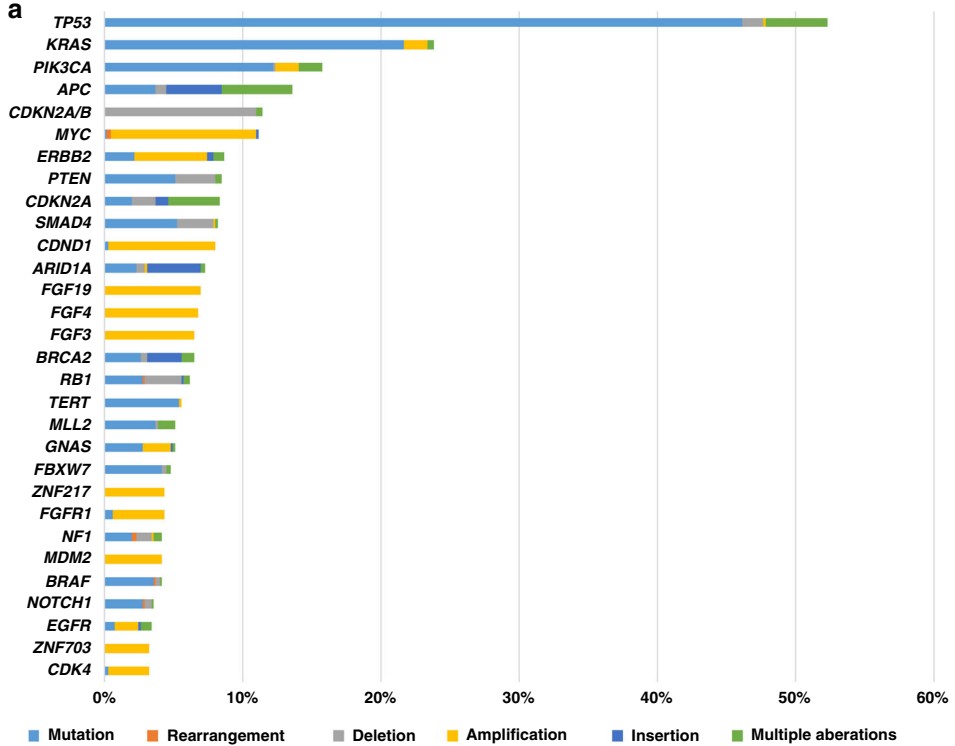

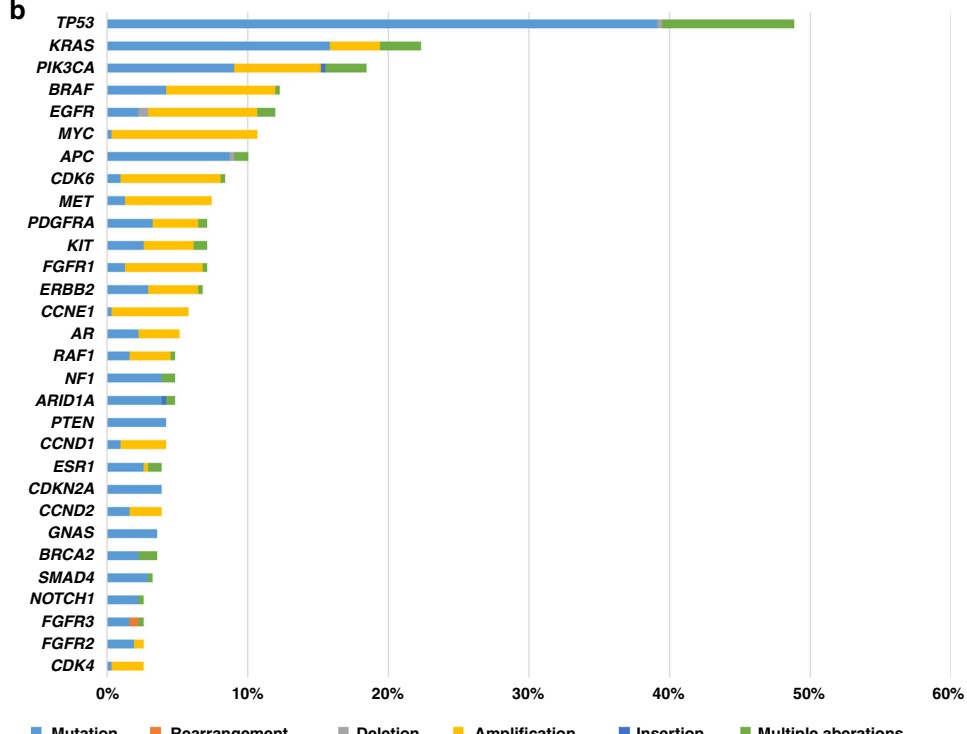

**Fig. 1 Frequency of characterized genomic alteration from tissue NGS and cfDNA. a** Frequency of characterized genomic alterations by tissue NGS ($N = 646$) (also see Supplementary Table 1). TP53 was the most commonly altered (52.3% [338/646]) followed by *KRAS* (23.8% [154/646]), *PIK3CA* (15.8% [102/646]), *APC* (13.6% [88/646]) and *CDKN2A/B* (11.5% [74/646]) alterations. *Percent indicates percent of the patient with alteration. Alterations seen in >3% of patients were included. **b** Frequency of characterized genomic alterations by cfDNA ($N = 309$) (See also Supplemental Table 2). The most common alterations were seen in TP53 (48.9% [151/309]) followed by *KRAS* (22.3% [69/309]), *PIK3CA* (18.4% [57/309]), *BRAF* (12.3% [38/309]) and *EGFR* (12.0% [37/309]) alterations. *Percent indicates percent of the patient with alteration. Alterations seen in >2.5% of patients were included.

(8/252). PD-L1 was positive in 18.5% (54/292) (defined as ≥1% positive by tumor proportion score or combined positivity score).

**Matching patients to drugs was feasible after MTB discussion.** Overall, 265 of 429 therapy evaluable patients (62%) were matched to at least one drug recommended by the MTB, including 86 of 429 patients (20%) matched to all recommended drugs including a combination therapy approach. The other patients (*N* = 164, 38%) received a physician's choice regimen (generally unmatched or low match) (Supplementary Methods and Supplementary Fig. 1).

**Compliance to MTB recommendations improved PFS and OS.** Patients who received the entire regimen recommended by the MTB had significantly improved PFS and OS when compared to patients who received physician's choice regimens (PFS: HR, 0.68; 95% CI, 0.51–0.90, *P* = 0.008, OS: HR, 0.69; 95% CI, 0.49–0.98, *P* = 0.036) (Fig. 2a, b) In contrast, patients who received part of the recommended MTB regimens had a trend towards improved PFS when compared to patients who received physician's choice regimen (HR,

0.85; 95% CI, 0.67–1.06, *P* = 0.153) (Fig. 2a) and there was no difference for OS between these two groups (HR, 0.97; 95% CI, 0.74–1.27, *P* = 0.815) (Fig. 2b) (univariate analysis). Of note, more than half (55.8% [48/86]) of the patients who received all the MTB-recommended medications received therapy with a high (≥ 50%) matching score while the majority of patients who received physician's choice regimen received therapy with low (<50%) matching score (95.7% [157/164]) (Supplementary Table 6).

**High matching score lead to better clinical outcomes.** We first allocated patients into four groups according to the matching score (Group A [*N* = 47]: matching score: 75–100%, B [*N* = 78]: 50–74%, C [*N* = 71]: 25–49%, D [*N* = 233]: 0–24%) and evaluated PFS and OS. Significantly prolonged PFS was observed among Groups A and B (high matching score groups) when compared to Group D (lowest matching score group) (A vs. D: HR, 0.47; 95% CI, 0.32–0.69; *P* = < 0.001, B vs. D: HR, 0.71; 95% CI, 0.54–0.94; *P* = 0.018) (Fig. 2c). There was no significant difference among lower matching score groups (C vs. D: HR,

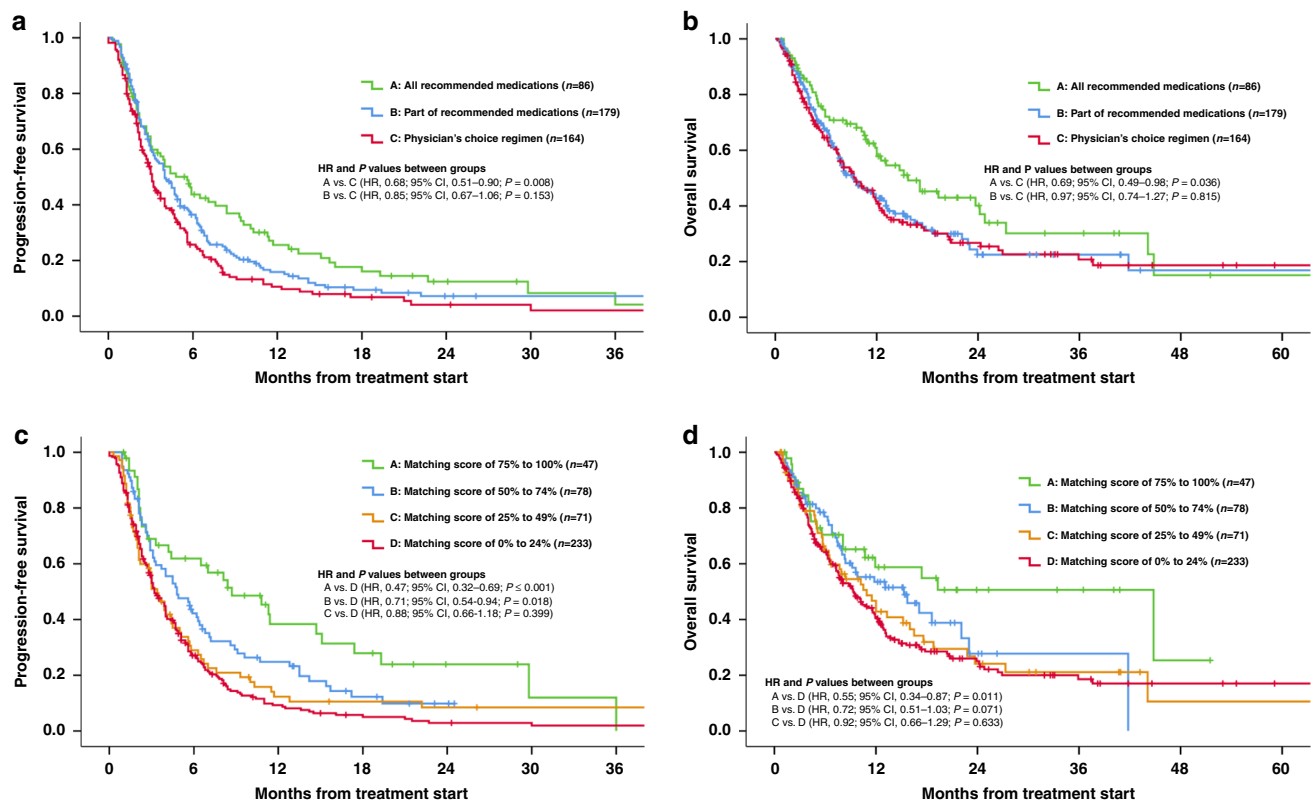

**Fig. 2 Progression-free survival and overall survival according to compliance with recommendation of Molecular Tumor Board and matching scores.**
**a** Progression-free survival according to compliance with recommendation of the Molecular Tumor Board (*N* = 429). Progression-free survival was significantly longer when Molecular Tumor Board recommendations were followed in full (group A) when compared to patients who received physician choice regimen (group C) (A vs. C: HR, 0.68; 95% CI, 0.51–0.90; *P* = 0.008, A vs. B: HR, 0.80; 95% CI, 0.60–1.06; *P* = 0.122, B vs. C: HR, 0.85; 95% CI, 0.67–1.06; *P* = 0.153). **b** Overall survival according to compliance with recommendation of Molecular Tumor Board (*N* = 429). Overall survival was significantly longer when Molecular Tumor Board recommendations were followed in full (group A) when compared to patients treated with physician choice regimen (group C). (A vs. C: HR, 0.69; 95% CI, 0.49–0.98; *P* = 0.036, A vs. B: HR, 0.72; 95% CI, 0.51–1.01; *P* = 0.056, B vs. C: HR, 0.97; 95% CI, 0.74–1.27; *P* = 0.815). **c** Progression-free survival according to matching scores (*N* = 429). Progression-free survival was longer among higher Matching Score groups (groups A and B) when compared to lower Matching Score patients (Group D) (A vs. D: HR, 0.47; 95% CI, 0.32–0.69; *P* < 0.001, B vs. D: HR, 0.71; 95% CI, 0.54–0.94; *P* = 0.018). Similarly, Group A versus C differed significantly, with Group A doing better (A vs. C: HR, 0.53; 95% CI, 0.34–0.82; *P* = 0.004) and there was a trend for Group A to be better than Group B (A vs. B: HR, 0.66; 95% CI, 0.43–1.01; *P* = 0.057). There was no significant difference between Group B vs. C (B vs. C: HR, 0.81; 95% CI, 0.57–1.15; *P* = 0.23). **d** Overall survival according to the matching scores (*N* = 429). Overall survival was longer among higher Matching Score groups (Group A) when compared to lower Matching Score (Group D) and trended longer for Group B vs. D and for Group A vs. C (A vs. D: HR, 0.55; 95% CI, 0.34–0.87; *P* = 0.011, B vs. D: HR, 0.72; 95% CI, 0.51–1.03; *P* = 0.071, A vs. C: HR, 0.59; 95% CI, 0.35–1.01; *P* = 0.053). There was however no difference between survival in groups A vs. B and B vs. C (A vs. B: HR, 0.76; 95% CI, 0.44–1.30; *P* = 0.312. B vs. C: HR, 0.79; 95% CI, 0.51–1.21; *P* = 0.269).

0.88; 95% CI, 0.66–1.18; $P = 0.399$) (Fig. 2c). High matching score groups also had a longer OS (A vs. D: HR, 0.55; 95% CI, 0.34–0.87; $P = 0.011$, B vs. D: HR, 0.72; 95% CI, 0.51–1.03; $P = 0.071$ [trend]) (Fig. 2d). Meanwhile, no difference in OS was seen among low matching score groups (C vs. D: HR, 0.92; 95% CI, 0.66–1.29; $P = 0.633$) (Fig. 2d) ($P$-values in univariate analysis).

Next, we stratified the patients according to matching scores ≥50% versus <50%, similar to our prior report[15] (Tables 2 and 3). Patients who received therapies with high ($N = 125$) versus low ($N = 304$) matching scores had significantly prolonged PFS and OS. (PFS: HR, 0.63; 95% CI, 0.50–0.80; $P < 0.001$; OS: HR, 0.67; 95% CI, 0.50–0.89; $P = 0.006$ [univariate]) (Table 2, Fig. 3a, b). Higher rates of SD ≥ 6 months/PR/CR were also seen among patients with high versus low matching score (52.1% vs. 30.3%, odds ratio [OR], 0.40; 95% CI, 0.26–0.62; $P < 0.001$ [univariate]) (Table 3 and Fig. 3c). Association between high matching score and improved PFS, OS and rate of SD ≥ 6 months/PR/CR remained significant after multivariate analysis (PFS: HR, 0.62; 95% CI, 0.47–0.81; $P = 0.001$; OS: HR, 0.67; 95% CI, 0.50–0.90; $P = 0.007$; rate of SD ≥ 6 months/PR/CR: OR, 0.40; 95% CI, 0.24–0.67; $P < 0.001$) (Tables 2 and 3).

## Discussion

Along with advances in molecular profiling technologies, targeted therapy and checkpoint blockade have revolutionized outcomes for some patients. However, treatment response can be short lived and substantial numbers of patients have primary or acquired resistance[1–5,20]. Although a number of basket and umbrella trials have been developed[21,22], challenges include low patient enrollment, which may be related to strict eligibility criteria. With the goal of improving patient management, multiple institutions are now implementing MTBs[17–19,23–26]. MTBs can facilitate clinical trial enrollment and following the MTBs recommendation may improve outcomes[23–26].

The MTB experience at the UCSD Moores Cancer Center for Personalized Therapy demonstrated that evaluation of different clinical-grade testing modalities (including tissue NGS, cfDNA, mRNA and IHC [including immune and chemotherapy biomarkers]) was feasible and facilitated the MTB discussion (Supplementary Table 1). Types and frequencies of genomic alterations identified were similar to previous reports (Fig. 1, Supplementary Tables 1 and 2)[17,18]. Overall, 265 of 429 evaluable patients (62%) were matched to ≥1 drug recommended by the MTB, and 86/429 patients (20%) were matched to all drugs recommended by the MTB, including combination approaches. The other patients ($N = 164$, 38%) received physician's choice regimen (usually unmatched or low degrees of matching) (Supplementary Fig. 1).

Physicians were permitted to choose which therapy they considered best for their patients regardless of MTB discussion, which was considered advisory. Similar to the previous report by Hoefflin et al.[23], patients who received an MTB-recommended regimen had significantly improved PFS and OS when compared to patients who received a physician's choice regimen (Fig. 2). Patients whose physicians adhered to MTB recommendations were more likely to receive matched targeted therapies that covered a larger fraction of their tumor's molecular alterations, yielding a high matching score, which may explain the improved clinical outcome (frequency of patients treated with ≥50% matching score: 55.8% [48/86] among patients who received all the MTB-recommended regimen vs. 4.3% [7/164] among patients who received physician's choice regimen [$P < 0.0001$]) (Supplementary Table 6). Consistent with this notion, significant improvement in PFS and OS were observed along with the step-wise increase in matching score (Fig. 3). Notably, patients who received therapy with high (≥50%) matching score

**Table 2 Association between demographic characteristics, PFS and OS ($N = 429$).**

| Characteristics | | PFS | | | | | | OS | | | | |
|---|---|---|---|---|---|---|---|---|---|---|---|---|
| | | | | Univariate | | Multivariate | | | Univariate | | Multivariate | |
| | | N | Median (months) (95% CI) | HR (95% CI) | P-value | HR (95% CI) | P-value | Median (months) (95% CI) | HR (95% CI) | P-value | HR (95% CI) | P-value |
| Age, years | ≥61 | 220 | 4.00 (2.98–5.02) | 0.89 (0.73–1.10) | 0.273 | — | — | 10.00 (7.61–12.39) | 1.06 (0.83–1.36) | 0.618 | — | — |
| | <61 | 209 | 4.00 (3.37–4.63) | — | — | — | — | 12.00 (10.13–13.87) | — | — | — | — |
| Sex | Male | 167 | 3.00 (2.30–3.70) | 1.12 (0.91–1.39) | 0.288 | — | — | 9.00 (6.13–11.88) | 1.16 (0.91–1.49) | 0.233 | — | — |
| | Female | 262 | 5.00 (4.32–5.68) | — | — | — | — | 12.00 (10.46–13.54) | — | — | — | — |
| Matching score (%) | ≥50% | 125 | 6.00 (4.27–7.73) | 0.63 (0.50–0.80) | <0.001 | 0.62 (0.47–0.81) | 0.001 | 17.00 (9.73–24.27) | 0.67 (0.50–0.89) | 0.006 | 0.67 (0.50–0.90) | 0.007 |
| | <50% | 304 | 4.00 (3.52–4.49) | — | — | — | — | 10.00 (8.34–11.66) | — | — | — | — |
| GI malignancies[a] | Yes | 123 | 4.00 (3.23–4.77) | 1.11 (0.88–1.39) | 0.381 | — | — | 8.00 (5.58–10.42) | 1.32 (1.01–1.72) | 0.041 | 1.32 (1.01–1.72) | 0.044 |
| | No | 306 | 4.00 (3.19–4.81) | — | — | — | — | 12.00 (10.13–13.87) | — | — | — | — |
| Number of prior lines of therapy | ≥3 | 204 | 4.00 (3.45–4.55) | 1.28 (1.04–1.57) | 0.021 | 1.27 (1.03–1.57) | 0.023 | 9.00 (6.47–11.53) | 1.48 (1.16–1.88) | 0.002 | 1.49 (1.16–1.90) | 0.001 |
| | <3 | 225 | 5.00 (4.09–5.91) | — | — | — | — | 13.00 (8.15–17.85) | — | — | — | — |
| Immunotherapy-based regimen[b] | Yes | 80 | 5.00 (2.49–7.51) | 0.79 (0.60–1.04) | 0.098 | 1.04 (0.76–1.42) | 0.829 | 13.00 (6.12–19.89) | 0.93 (0.67–1.29) | 0.651 | — | — |
| | No | 349 | 4.00 (3.35–4.65) | — | — | — | — | 11.00 (9.25–12.75) | — | — | — | — |

[a]Among 123 patients with heterogeneous GI malignancies, matching score of ≥50% ($N = 33$) vs. <50% ($N = 90$) showed significant difference in PFS (HR: 0.62, 95% CI: 0.39–0.98, $P = 0.042$). However, there was no difference in OS (HR: 0.74, 95% CI: 0.43–127, $P = 0.268$.
[b]Sixty of the 80 patients receiving immunotherapy were matched based on profiling.
CI confidence interval, HR hazard ratio, GI gastrointestinal, OS overall survival, PFS progression-free survival.

**Table 3 Association between patient and treatment characteristics and clinical benefit rate (SD ≥ 6 months/PR/CR) (N = 405\*).**

| Characteristics | | | Clinical benefit rate (SD ≥ 6 months/PR/CR) | | | | |
|---|---|---|---|---|---|---|---|
| | | | | Univariate | | Multivariate | |
| | | N | SD ≥ 6 months/PR/CR (N, %) | OR (95% CI) | P-value | OR (95% CI) | P-value |
| Age, years | ≥61 | 207 | 83 (40.1%) | 0.75 (0.50-1.12) | 0.159 | 0.77 (0.51-1.17) | 0.222 |
| | <61 | 198 | 66 (33.3%) | — | — | — | — |
| Sex | Male | 157 | 54 (34.4%) | 1.18 (0.78-1.80) | 0.427 | — | — |
| | Female | 248 | 95 (38.3%) | — | — | — | — |
| Matching score (%) | ≥50% | 119 | 62 (52.1%) | 0.40 (0.26-0.62) | <0.001 | 0.40 (0.24-0.67) | <0.001 |
| | <50% | 286 | 87 (30.4%) | — | — | — | — |
| GI malignancies | Yes | 114 | 35 (30.7%) | 1.45 (0.92-2.31) | 0.113 | — | — |
| | No | 291 | 114 (39.2%) | — | — | — | — |
| Number of prior lines of therapy | ≥3 | 194 | 60 (30.9%) | 1.63 (1.08-2.45) | 0.019 | 1.51 (1.00-2.30) | 0.052 |
| | <3 | 211 | 89 (42.2%) | — | — | — | — |
| Immunotherapy-based regimen | Yes | 77 | 35 (45.5%) | 0.64 (0.39-1.06) | 0.081 | 1.12 (0.62-2.03) | 0.709 |
| | No | 328 | 114 (34.8%) | — | — | — | — |

*Twenty-four of 429 patients were not evaluable for response since these patients had ongoing SD that was less than 6 months at the time of data cutoff.
CI confidence interval, CR complete response, GI gastrointestinal, OR odds ratio, PR partial response, SD stable disease.

demonstrated significant improvement in PFS and OS when compared to patients who were treated with low matching score (<50%) (PFS: HR, 0.62; 95% CI, 0.47–0.81; $P = 0.001$, OS: HR, 0.67; 95% CI, 0.50–0.90; $P = 0.007$ [multivariate]). Rate of clinical benefit (SD ≥ 6 months/PR/CR) was also significantly higher among patients whose therapy had a high (≥50%) matching score when compared to the low (<50%) matching score group (SD ≥ 6 months/PR/CR: 52.1% vs. 30.3%, OR, 0.40; 95% CI, 0.24–0.67; $P < 0.001$ [multivariate]) (Table 2 and Fig. 3c). Overall, our MTB experience suggests that greater degrees of matching of tumors to drugs, including with customized N-of-one recommended combinations, was independently associated with better outcomes.

These results are consistent with previous preliminary work from our group[27] as well as from our prospective trial–Investigation of Profile-Related Evidence Determining Individualized Cancer Therapy (I-PREDICT) [NCT02534675]) trial, where 73 patients with treatment-refractory solid tumors were treated with a combination-based strategy using their unique tumor genomic signatures[15]. Patients whose genomic aberrations were targeted with high matching scores demonstrated significantly better clinical outcomes including response rate, PFS, OS when compared to the lower matching group. Similar results were seen in the WINTHER trial (NCT01856296) where patients were navigated to therapy on the basis of DNA as well as RNA profiling[16]. Further investigations with larger sample size are required to determine if certain combination approaches are more efficacious than others. Of note, in the current study, there were 10 patients who had single gene alterations that were matched with targeted therapies, and 8/10 such patients (80%) attained clinical benefit including $N = 5$ partial responses (PFS: 8, 11 +, 19, 20+ and 36 months) and $N = 3$ durable stable disease ≥6 months (PFS: 11 +, 15 and 22+ months) (with "+" indicating ongoing responses). Hence, single gene targeting may be sufficient to achieve clinical response among patients harboring a limited number of genomic alterations.

There are several obstacles to the implementation of MTBs: (i) MTBs require multidisciplinary expertise; smaller practices may benefit from collaborations by remote video conference/virtual tumor boards[28]; (ii) lack of access to drugs or clinical trials that limits patients being treated with the recommended regimen; clinical trials need to be available locally and expanded use of existing anticancer therapies is needed as leveraged by the Drug Rediscovery protocol (DRUP) developed in the Netherlands[29] and the PREDICT[18] and I-PREDICT protocols[15]; (iii) heterogeneity of MTB recommendations across different institutions can also be challenging[25]; sharing clinical

data experiences between institutions and expert decision support could attenuate this issue; (iv) the field of molecular oncology is rapidly evolving, and hence the understanding and interpretation of biomarkers and outcomes can change. Indeed in this study, we have observed an increase in matching score over time (frequency of matching score ≥50%: December 2012–December 2015, 15.3% [23/150] vs. January 2016–September 2018, 36.6% [102/279]).

There were several limitations to our current study. First, it was not a randomized controlled trial but rather reflects a real-world experience. Second, the number of cancer types included in the study was based on the treating physicians who requested MTB discussion, thereby predisposing to selection bias. Third, molecular analysis was obtained at various time points in relationship to the clinical history. Lastly, differences in molecular profiling platforms may have affected the detection of targetable markers.

In summary, our MTB successfully facilitated the interpretation of multiple testing modalities including tissue NGS, cfDNA, mRNA and IHC. Patients whose physicians followed the MTB discussion recommendations received therapy that was better matched to their alterations and achieved significantly better clinical outcomes, including longer PFS and OS when compared to patients who received physician's choice regimen. Moreover, patients who received treatment that matched a larger fraction of identified molecular anomalies (i.e., had higher matching scores) did better, and the degree of matching was an independent predictor of improvement in all outcome parameters, including OS, in multivariate analysis. These data are consistent with previous observations indicating that patients with advanced/metastatic disease often have complex and distinct molecular alterations[11,12,14,30] that require N-of-One matched treatments[15], rather than standard monotherapies or unmatched therapy based on non-biomarker based population trials. Further clinical investigation is warranted in order to validate these findings, as well as to determine if there are matching score thresholds that determine the utility of precision therapies.

## Methods

**Molecular tumor board.** The molecular tumor board (MTB) (face-to-face) meetings were held for 1–1.5 h, ~3 times/month at the UCSD Moores Cancer Center. Cases to be discussed were submitted by the treating physicians. A handout prepared by the MTB project manager included a meeting agenda, de-identified patient information (age and sex, physician's name, diagnosis and date of diagnosis, last treatment, biopsy site and date, molecular test used, molecular profile results, and comments), and a copy of the key parts of the molecular diagnostic report. The manager also facilitated screening laboratories for certification and assisted physicians with ordering tests and helped with obtaining consent when requested.

A senior and a mid-level medical oncologist experienced in clinical trials, genomics, and immunotherapy moderated the meeting. The MTB attendees included

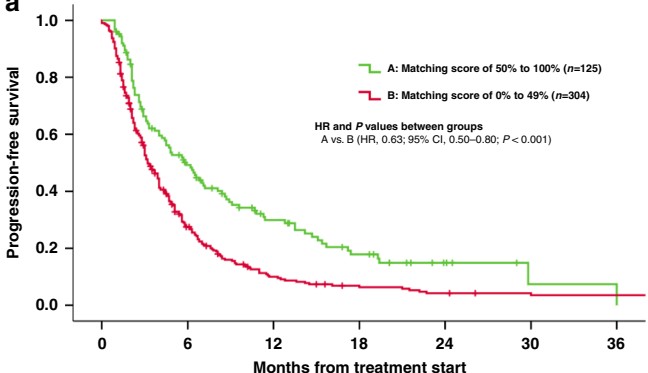

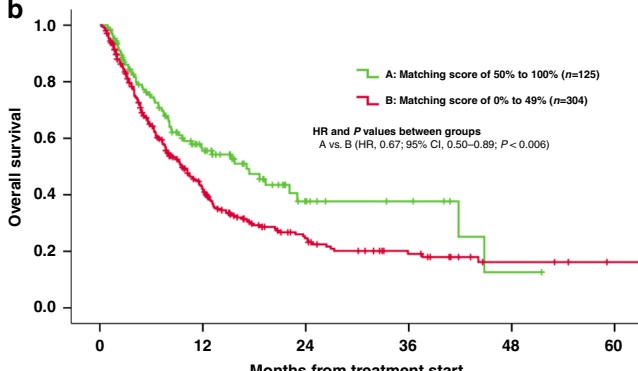

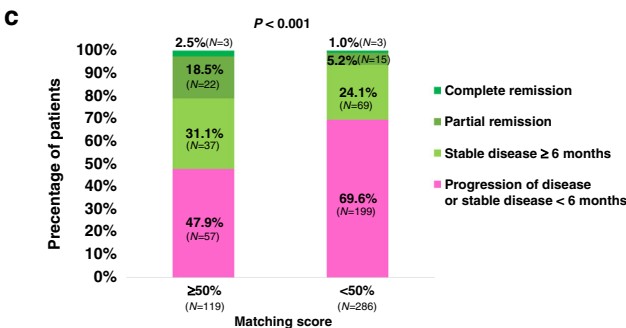

**Fig. 3 Progression-free survival, overall survival and clinical benefit rate (SD≥ 6 months/PR/CR) according to the high (≥50%) vs. low (<50%) matching scores. a** Progression-free survival according to the high (≥50%) vs. low (<50%) matching scores ($N = 429$). Progression-free survival was significantly longer among patients who received therapy with high (≥50%) matching score when compared to patients treated with low (<50%) matching scores. **b** Overall survival according to the high (≥50%) vs. low (<50%) matching scores ($N = 429$). Overall survival was significantly longer among patients who received therapy with high (≥50%) Matching Score when compared to patients treated with low (<50%) Matching Scores. **c** Clinical benefit rate (SD ≥ 6 months/PR/CR) according to the high (≥50%) vs. low (<50%) matching scores ($N = 405*$). Clinical benefit rate (SD ≥ 6 months/PR/CR) was significantly higher among patients who received therapy with high (≥50%) Matching Score when compared to patients treated with low (<50%) Matching Scores (52.1% vs. 30.3% [$P < 0.001$]).

medical oncologists, surgeons, radiation oncologists, clinical trial coordinators/navigators, geneticists, pathologists, radiologists, gynecologic oncologists, basic/translational scientists and bioinformaticians. Clinical laboratory improvement amendments (CLIA)-licensed and College of American Pathologist (CAP)-accredited clinical laboratory tests were evaluated. Imaging and pathology were reviewed along with clinical history. Discussion concentrated on the impact of aberrations on signaling pathways, whether germline aberrations might also be present; and which

drugs, either approved or in clinical trials, might impact the molecular alterations. A project manager assisted with test ordering and answering physician and patient questions and consent, hence facilitating the process. A medication acquisition specialist and clinical trial coordinators/navigators attended the MTB to assist in obtaining medications (either off- or on-label approved), and screened for available clinical trials. MTB adhered to all HIPAA privacy laws. An MTB moderator and the presenting physician evaluated the accuracy of the review before documentation in the medical record. The MTB recommendations were considered advisory, with the treating physician making final decisions with respect to therapy.

**Patients**. In the current study, electronic medical records were reviewed for patients' characteristics and outcome, for individuals presented at the MTB between December 2012 and September 2018. This study followed the guidelines of the IRB-approved UCSD- Profile Related Evidence Determining Individualized Cancer Therapy (PREDICT) study (NCT02478931, https://clinicaltrials.gov/ct2/show/NCT02478931) and any investigational studies for which the patients gave consent. (Details of included/excluded patients are shown in Supplementary Fig. 1.)

**NGS of tissue and cell-free circulating tumor DNA (cfDNA)**. Tissue and blood NGS was conducted in one of several CLIA certified laboratories (see Supplementary Table 1) (182 to 596 genes in tissue panels and 54 to 74 genes for blood-derived cfDNA), depending on the laboratory and time frame. mRNA and protein expression analysis (including for immune markers) were also evaluated in selected patients (Supplementary Table 1). The treating physician determined the choice of test.

**Statistical methods**. Patient and molecular characteristics were summarized by descriptive statistics. We assessed progression-free survival (PFS), which was defined as time between start of the treatment after the MTB and the date of progression confirmed by imaging or clinical findings. Overall survival (OS) was defined as time between start of the therapy after MTB presentation until last follow-up. Patients with ongoing therapy without progression at the last follow-up date were censored for PFS at that date. Patients alive at last follow-up were censored for OS. Response was assessed in accordance with RECIST criteria[31]. Log-rank test and Kaplan–Meier analysis were used to compare subgroups of patients. $P$-values ≤0.05 were considered significant. Statistical analyses were performed by HK with SPSS, version 25 (IBM Corporation, Armonk, NY).

**Matching score**. All NGS pathogenic variants (but no variants of unknown significance [VUS]) were included in the matching score calculation as previously described[15]. However, protein or RNA were only included in the calculation when they were targeted.

Briefly, the matching score evaluated the number of pathogenic alterations targeted by drugs given divided by total number of pathogenic alterations: the higher the score, the better the match (0%, unmatched; 100%, completely matched). For example, if a tumor harbored eight genomic alterations and the patient received two agents that targeted four of these alterations, the score would be 50% (4 of 8). Investigators blinded to patient outcomes determined the scores. Since there can be heterogeneity between blood and tissue samples or between two tissue biopsies, if a patient had more than one NGS or other biomarker report, the alterations in each report were counted; however, if the two reports were from the same laboratory and of the same type (but from a different blood or tissue sample), the sample closest to the time of MTB was used.

Additional details regarding matching score calculations are included in Supplementary Methods[32–39]. Data for 429 patients who were evaluable for therapeutic outcome after MTB discussion are included in Supplementary Dataset 1.

**Reporting summary**. Further information on research design is available in the Nature Research Reporting Summary linked to this article.

## Data availability

All data generated or analyzed during this study are included in this published article (and its supplementary information files).

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

## Acknowledgements

This work was supported in part by the Joan and Irwin Jacobs Fund and by National Cancer Institute at the National Institutes of Health [Grant No. NIH P30 CA023100 (RK)].

## Author contributions

S.K., K.H.K., J.J.A. and R.K. drafted the manuscript; S.K. and R.K. designed the study; S.K., K.H.K. and R.O. analyzed the data; K.H.K., H.J.L., R.O. and S.L. collected the data. S.K., A.B., M.N., E.W., D.J.K., R.N.E., A.G., N.G., P.T.F., R.B.S., R.S., S.C.P., A.S., E.S., R.O., S. Lee, S.M.L., J.K.S. and R.K. were involved in molecular tumor board. All authors have read and approved the final manuscript.

## Competing interests

S.K. serves as a consultant for Foundation Medicine and receives speaker's fees from Roche. R.E. serves as a consultant for GSK, Merck, Eisai, Clovis Oncology, AstraZenica, Pfizer and Iovance. He has received speaker fees from AstraZenica and Merck. A.S. reports research funding and honoraria from Pfizer and Varian Medical Systems, consultant fees from AstraZeneca and Jounce Therapeutics, personal fees from Merck, and is the scientific founder with equity interest in Toragen Inc. outside the submitted work. J.K.S. receives research funds from Foundation Medicine Inc. and Amgen, as well as consultant fees from Grand Rounds, Deciphera and LOXO. R.K. has research funding from Incyte, Genentech, Merck Serono, Pfizer, Sequenom, Foundation Medicine, Guardant Health, Grifols, and Konica Minolta, as well as consultant fees from LOXO, X-Biotech, Actuate Therapeutics, Genentec, Pfizer and NeoMed. She receives speaker fees from Roche, and has an equity interest in IDbyDNA and Curematch, Inc. She is a co-founder of CureMatch and Board member of CureMatch and CureMetrix. The remaining authors declare no competing interests.

## Additional information

Shumei Kato [1,10]✉, Ki Hwan Kim[1,2,10]✉, Hyo Jeong Lim[1,3], Amelie Boichard[1], Mina Nikanjam[1], Elizabeth Weihe[4], Dennis J. Kuo [5], Ramez N. Eskander[1], Aaron Goodman[1], Natalie Galanina[1], Paul T. Fanta[1], Richard B. Schwab[1], Rebecca Shatsky[1], Steven C. Plaxe[1], Andrew Sharabi[1,6], Edward Stites [7], Jacob J. Adashek [8], Ryosuke Okamura [1], Suzanna Lee[1], Scott M. Lippman[1], Jason K. Sicklick[9] & Razelle Kurzrock[1]

[1]Center for Personalized Cancer Therapy and Division of Hematology and Oncology, Department of Medicine, UC San Diego Moores Cancer Center, La Jolla, CA, USA. [2]Division of Hematology and Medical Oncology, Department of Internal Medicine, Seoul National University Boramae Medical Center, Seoul, Republic of Korea. [3]Department of Internal Medicine, Veterans Health Service Medical Center, Seoul, Republic of Korea. [4]Department of Radiology, UC San Diego Moores Cancer Center, La Jolla, CA, USA. [5]Division of Pediatric Hematology-Oncology, Rady Children's Hospital-San Diego, University of California San Diego School of Medicine, San Diego, CA, USA. [6]Department of Radiation Medicine and Applied Sciences, UC San Diego Moores Cancer Center, La Jolla, CA, USA. [7]Integrative Biology Laboratory, Salk Institute for Biological Studies, La Jolla, CA, USA. [8]Department of Internal Medicine, University of South Florida, H. Lee Moffitt Cancer Center & Research Institute, Tampa, FL, USA. [9]Center for Personalized Cancer Therapy and Division of Surgical Oncology, Department of Surgery, UC San Diego Moores Cancer Center, La Jolla, CA, USA. [10]These authors contributed equally: Shumei Kato, Ki Hwan Kim. ✉email: smkato@ucsd.edu; floresta405@gmail.com

