## [Peer Review File · Nature Communications]

Reviewers' Comments:

Reviewer #1:

Remarks to the Author:

Dear authors,

I have reviewed your manuscript „Real-World Data from a Molecular Tumor Board: Improved Outcomes with a Precision N-of-One Strategy” where patient outcomes of a molecular tumor board were improved with better molecular matching scores.

I agree, that a focus on flexible N-of-One strategies is promising and required in precision oncology and the establishment of a matching score provides a potentially helpful tool for patient stratification. In this regard, I think that this is an interesting manuscript, describing a retrospective real-world validation cohort with an adequate number of patients and convincing results.

However, I have a few comments regarding the manuscript:

1. The authors have previously described results from the prospective I-PREDICT and WINTHER trials. It would be of interest to know if some of the here retrospectively described patients have been part of a previously published study or if they constitute an entirely novel validation cohort.
2. Matching genomic alterations to treatment options is not a simple task. I would be interested in an additional methods section on how treatment options were identified from genomic alterations (databases used if any, tumor type taken into account, clinical evidence necessary or also preclinical studies taken into account...).
3. It would be interesting to get more details of the individual patients on a case-report level, e.g. tumor type, molecular alterations, treatment and response. A communication of these individual data helps with the understanding of the workflow, as well as with learning from each N-of-One study. At least a list of recurrent therapeutic interventions for specific biomarkers (e.g. specific combinations of targeted therapy and/or chemotherapy) and their outcome could help with understanding workflow and potential application to other patient populations.
4. An underlying master protocol is briefly mentioned in abstract and introduction, some more details on this protocol should be provided.
5. The number of genomic alterations per patient is of interest to better understand the impact of complex genotypes on treatment and response (e.g. in Table 2). Was a higher matching score mainly achieved in patients where only one well-known targetable alteration was identified? Since the matching score is especially interesting for patients with genotypes that do not typically seem “targetable” with a single drug, a potential impact of more “simple” or “complex” genotypes should be discussed.
6. How many percent of patients received immune checkpoint inhibition? Was this independently associated with improved outcome? How often was immunotherapy guided by molecular alterations?
7. In table 2, GI malignancies are an independent predictor of worse survival. Could you discuss this finding with regard to precision oncology (e.g. do these malignancies still benefit from your PO strategy)?
8. You discuss promising results by Hoefflin et al. Yet, other PO studies have failed to reach prespecified endpoints, such as the WINTHER and SHIVA trials. I believe that a discussion of the matching score and customized combinations in light of these negative results would be interesting (e.g. should patients only receive personalized treatment if an adequate matching score can be achieved? Which prospective trials should follow these data?).

Minor comments:

1. Typo “under the auspices of master protocol” in the abstract
2. Typo “entretctinib” in line 4 of the introduction.
3. Typo “the MTB served as an advisory board, with final decision made by...” in the introduction
4. Typo “in certain circumstances, physician did not change therapy...” in results, patient characteristics.
5. Typo “primary or acquired resistant” in line 4 discussion.

6. Typo Caption Figure 2D "there were however no difference..."

7. "A project manager assisted with test ordering and answering physician and patient questions and consent, hence facilitating the process... and drug acquisition and clinical trial..." from the discussion section should rather go into methods.

Reviewer #2:

Remarks to the Author:

Kato et al's manuscript 'Real-World Data from a Molecular Tumor Board: Improved Outcomes with Precision N-of One Strategy' summarises the implementation and use of a molecular tumor board over a 6 year period. The group reviewed more than seven hundred patients and were able to 'match' >60% of those to at least one recommended drug. Although the report is very detailed and well written, several groups have recently published their MTB experience with similar outcomes.

One of the novelties the group is reporting is a proposed matching score (in the supplemental part) which can identify patients with better response and outcome. Although this tool may predict response it is not at all validated or tested in a larger cohort of patients. The subsequent recommendation of applying matching drugs at 50% (2 drugs) or 33% (3 drugs) of their standard dose is questionable and not standard practice.

Reviewer #3:

Remarks to the Author:

In this manuscript, the authors describe the experiences and associated patient outcomes from a precision medicine effort implemented by a multidisciplinary molecular tumor board (MTB). The work is an extension (both with respect to # of patients and analytes assessed) of the I-PREDICT study (Nat. Med., 2019). In short, the data indicates that patients who received a MTB-recommended regimen had significantly improved outcome (PFS, OS and clinical benefit rate). The degree of matching alterations (defined as a Matching Score) similarly associated with improved outcome. As the authors make clear in the discussion, this is not a randomized controlled study and could be influenced by potential selection bias; hence making it difficult to fully conclude that the benefit observed was specifically due to the MTB-recommended regimen administered. Although this needs to be taken into account, these are inherent challenges in assessing real-world clinical datasets of limited size.

The authors note that the results in this manuscript are consistent with prior analyses from the I-PREDICT trial where 73 patients received similar MTB-recommended regimens. However, there was one potentially notable difference in objective response (which is likely the best indicator whether the MTB-recommended regimen was active). In I-PREDICT, 45% of patients with $\geq 50\%$ matching scores achieved PR/CR, whereas 21% of patients achieved PR/CR in this manuscript. (the overall clinical benefit rate does not differ greatly due to the fact that only 5% patients had SD ≥ 6 months in I-PREDICT, versus 31% in this report). Given that there appeared to be a broader range of biomarkers (notably IHC and mRNA highlighted Supplemental Tables 4 & 5) that contributed to treatment recommendations, it would be useful for the authors to comment on how the types of biomarkers may have influenced the outcome. (or were there other factors that influenced the lower PR/CR rate in this report compared to the initial I-PREDICT data?) For instance, was PR/CR or clinical benefit rate in the $\geq 50\%$ matching score group driven primarily by validated, genomically-defined targeted agents? How often were the less validated IHC tests used to guide a treatment recommendation and did they provide any value with respect to outcome? Compared to other reported MTB-guided precision medicine efforts, this study appears to represent one of the larger based upon the # of patients evaluable for therapy after MTB decisions. As such, the authors should comment on their learnings regarding the value of specific

biomarkers. Nevertheless, the authors should at least provide some granularity around treatment recommendations, as this was not noted in the paper. A table similar to Supplementary Table 2 in their Nature Medicine 2019 paper would be informative to the reader (including Matching Score).

Given that a majority of the patients who received physician's choice regimens had matching scores <50% (ie. 95% of the 164 patients), was this associated with an earlier review during the 2012-2018 timeframe? (Put another way, the degree of biomarker content and more importantly the degree of confidence in the interpretation of biomarker/outcome associations would have been limited in the earlier years of this study, potentially leading to physician's not following MTB recommendations)

What is the criteria for IHC/mRNA positivity? Given that it was utilized for treatment recommendations, it should be indicated.

The authors should incorporate the number of centers patients treated who contributed to this study?

The prevalence of specific genomic aberrations was higher in the cfDNA dataset compared to the tissue NGS dataset. Of note, amplifications of BRAF, EGFR, CDK6, MET (as examples) approached ~7% in cfDNA, but were not identified in tissue DNA. There could be several explanations, including timepoint of sample acquisition in relationship to clinical treatment or technical given the differences in platforms. Since there are potentially actionable therapies for these specific examples, it would be informative to understand the criteria for determining amplification of genes on the panel if they were used for treatment recommendations.

REVIEWER COMMENTS

Reviewer #1 (Remarks to the Author):

Dear authors,

I have reviewed your manuscript „Real-World Data from a Molecular Tumor Board: Improved Outcomes with a Precision N-of-One Strategy” where patient outcomes of a molecular tumor board were improved with better molecular matching scores.

I agree, that a focus on flexible N-of-One strategies is promising and required in precision oncology and the establishment of a matching score provides a potentially helpful tool for patient stratification. In this regard, I think that this is an interesting manuscript, describing a retrospective real-world validation cohort with an adequate number of patients and convincing results.

Answer:

We thank the reviewer’s favorable comments.

However, I have a few comments regarding the manuscript:

1. The authors have previously described results from the prospective I-PREDICT and WINTHER trials. It would be of interest to know if some of the here retrospectively described patients have been part of a previously published study or if they constitute an entirely novel validation cohort.

Answer:

There was no overlap with the WINTHER study.

Of 265 patients with matched therapy in the current manuscript, 27 were part of the IPREDICT cohort reported. This is now stated in footnote to Supplemental Figure 1. The reason that there was such a small overlap was that the IPREDICT study often used an electronic molecular tumor board, while the current study was strictly derived from the face-to-face molecular tumor board, as now stated in the Methods.

2. Matching genomic alterations to treatment options is not a simple task. I would be interested in an additional methods section on how treatment options were identified from genomic alterations (databases used if any, tumor type taken into account, clinical evidence necessary or also preclinical studies taken into account...).

Answer:

We agree that matching genomic alterations to treatment is an evolving science. This is why our molecular tumor board was multidisciplinary and included medical oncologists, surgeons, radiation oncologists, clinical trial coordinators/navigators, geneticists,

pathologists, radiologists, gynecologic oncologists, basic/translational scientists and bioinformaticians, as stated in the Methods. We took into an account both clinical and pre-clinical evidence, though of course clinical evidence would be more influential. Finally, most of the testing was done through Foundation Medicine. The Foundation Medicine database is very large and their readings (which includes drug suggestions) are now FDA approved. The Methods and supplemental Methods provide details on how we assessed the matching.

3. It would be interesting to get more details of the individual patients on a case-report level, e.g. tumor type, molecular alterations, treatment and response. A communication of these individual data helps with the understanding of the workflow, as well as with learning from each N-of-One study. At least a list of recurrent therapeutic interventions for specific biomarkers (e.g. specific combinations of targeted therapy and/or chemotherapy) and their outcome could help with understanding workflow and potential application to other patient populations.

Answer:

We have now provided a database of all 429 patients evaluable for clinical outcome that includes patient demographics, cancer diagnosis, molecular information, treatment regimen, rate of matching, progression-free survival, overall survival and clinical benefit. We believe this information will be helpful to advance the N-of-One approach (provided as Supplemental Data).

4. An underlying master protocol is briefly mentioned in abstract and introduction, some more details on this protocol should be provided.

Answer:

We have now included the online link in the Patients and Methods section, Page 17. The protocol was a master observational study that permitted data collection on these patients. If needed, we can provide the protocol.

5. The number of genomic alterations per patient is of interest to better understand the impact of complex genotypes on treatment and response (e.g. in Table 2). Was a higher matching score mainly achieved in patients where only one well-known targetable alteration was identified? Since the matching score is especially interesting for patients with genotypes that do not typically seem “targetable” with a single drug, a potential impact of more “simple” or “complex” genotypes should be discussed.

Answer:

We thank reviewer for this insightful question.

Of the 47 patients with high Matching Score of 75% to 100%, only 10 patients had a single alteration that was matched to the targeted therapies (giving them a matching score of 100%)

Among those 10 patients, five achieved a partial response including durable responses (PFS: 8, 11+, 19, 20+ and 36 months), and 3 additional patients had stable disease \geq 6 months (PFS: 11+, 15 and 22+ months) (with “+” sign indicating ongoing responses). Although this is small number of patients, we believe this is worth highlighting in the discussion.

We have now commented in the Discussion (Page 13) as following:

“Of note, in this study, there were 10 patients who had a single gene alteration that was matched with targeted therapies and 8/10 (80%) patients had clinical benefit including N=5 partial response (PFS: 8, 11+, 19, 20+ and 36 months) and N=3 durable stable disease \geq 6 months (PFS: 11+, 15 and 22+ months) (with “+” indicating ongoing responses).”

6. How many percent of patients received immune checkpoint inhibition? Was this independently associated with improved outcome? How often was immunotherapy guided by molecular alterations?

Answer: We thank reviewer for this excellent question.

Eighty patients received immunotherapy and most cases (N=60) were given immunotherapy based on the molecular profiling (e.g., microsatellite instability, high tumor mutational burden etc).

We have re-analyzed the data based on treatment with immunotherapy-based regimens and found that immunotherapy had favorable trend (albeit not statistically significant) for PFS (HR: 0.79, P=0.098) and clinical benefit rate (OR: 0.64, P=0.081) by univariate analysis. It was not statistically significant after the multivariate analysis. We did not see a difference in OS based on immunotherapy. We have now updated these results in Tables 2 and 3. Multivariate analysis was also updated (which did not change the overall outcome). These results indicate that immunotherapy in of itself did not predict outcome; rather it was the degree of matching as reflect by matching score that was important in multivariate analysis.

7. In table 2, GI malignancies are an independent predictor of worse survival. Could you discuss this finding with regard to precision oncology (e.g. do these malignancies still benefit from your PO strategy).

Answer:

Among 123 patients with heterogeneous GI malignancies, Matching Score of \geq 50% (N=33) vs. <50% (N=90) showed significant difference in PFS (HR: 0.62, 95% CI: 0.39-0.98,

P = 0.042). However, there was no difference in OS (HR: 0.74, 95% C.I: 0.43-1.27, P = 0.268). We believe there is some signal for PFS benefit with matched targeted therapy approach among GI cancers, but the number is too small for any conclusion, especially since the GI malignancies were heterogeneous with multiple different tumor types. Further investigation is necessary.

We have added the above information as a footnote to Table 2.

8. You discuss promising results by Hoefflin et al. Yet, other PO studies have failed to reach prespecified endpoints, such as the WINTHER and SHIVA trials. I believe that a discussion of the matching score and customized combinations in light of these negative results would be interesting (e.g. should patients only receive personalized treatment if an adequate matching score can be achieved? Which prospective trials should follow these data?).

Answer:

The reviewer correctly observed that the patient proportion with WINTHER versus previous therapy progression-free survival ratio of >1.5 was 22.4%, which did not meet the pre-specified primary end point. However, it should be noted that comparing trial PFS to prior PFS has important confounders, which we did not realize when the trial was first designed in 2012. For instance, imaging time points are often inconsistent in the prior PFS (since the patient is not on trial) and time between imaging can influence the prior PFS. Even so, matching score was significantly and independently associated with outcome (PFS and OS) in the WINTHER trial.

In regards to the SHIVA trial, it is an important study because it was the first randomized precision medicine trial. However, it was limited by the fact that a majority of patients in the matching arm received single agent mTOR inhibitor or hormonal modulators.

Based on these data alone, we do not believe that we are ready to claim that patients should only receive personalized treatment if an adequate matching score can be achieved. However, this may be a good question for future studies.

We have now added the statement at Discussion as following:

Further clinical investigation is warranted in order to validate these findings as well as to determine if there are Matching Score thresholds that determine the utility of precision therapies.

Minor comments:

1. Typo “under the auspices of master protocol” in the abstract

Answer: Corrected. Now says “under the auspices of a master protocol.”

2. Typo “entretctinib” in line 4 of the introduction.

Answer: We have now updated this.

3. Typo “the MTB served as an advisory board, with final decision made by...” in the introduction

Answer: Grammar corrected

4. Typo “in certain circumstances, physician did not change therapy...” in results, patient characteristics.

Answer: Wording corrected.

5. Typo “primary or acquired resistant” in line 4 discussion.

Answer: Grammar corrected.

6. Typo Caption Figure 2D “there were however no difference...”

Answer: Grammar corrected.

7. “A project manager assisted with test ordering and answering physician and patient questions and consent, hence facilitating the process... and drug acquisition and clinical trial...” from the discussion section should rather go into methods.

Answer: We have now moved this sentence to Patients and Methods.

Reviewer #2 (Remarks to the Author):

Kato et al's manuscript 'Real-World Data from a Molecular Tumor Board: Improved Outcomes with Precision N-of One Strategy' summarises the implementation and use of a molecular tumor board over a 6 year period. The group reviewed more than seven hundred patients and were able to 'match' >60% of those to at least one recommended drug. Although the report is very detailed and well written, several groups have recently published their MTB experience with similar outcomes.

One of the novelties the group is reporting is a proposed matching score(in the supplemental part)which can identify patients with better response and outcome. Although this tool may predict response it is not at all validated or tested in a larger cohort of patients. The subsequent recommendation of applying matching drugs at 50% (2 drugs) or 33% (3drugs) of their standard dose is questionable and not standard practice.

Answer:

We thank the reviewer for his/her comment stating that “one of the novelties” in this study is the Matching Score.

We agree that further validation of the Matching Score in prospective and larger studies is required. We have now commented in Discussion, Pages 14-15 as following:

“Further clinical investigation is warranted in order to validate these findings as well as to determine if there are Matching Score thresholds that determine the utility of precision therapies.”

Regarding the dosing, when we gave novel combinations, we lowered the dose, as noted by the reviewer in order to assure safety. However, the doses were then escalated to tolerance in each individual patient.

We say: “Patients were monitored closely by the treating physician, and if the combination therapies were tolerable, dosage was escalated to tolerance.”

This schema for dosing combinations was derived from analysis of ~70,000 patients in the literature (references 37-39) and has been shown to be both safe and effective in our prior study (reference 15).

Reviewer #3 (Remarks to the Author):

In this manuscript, the authors describe the experiences and associated patient outcomes from a precision medicine effort implemented by a multidisciplinary molecular tumor board (MTB). The work is an extension (both with respect to # of patients and analytes assessed) of the I-PREDICT study (Nat. Med., 2019). In short, the data indicates that patients who received a MTB-recommended regimen had significantly improved outcome (PFS, OS and clinical benefit rate). The degree of matching alterations (defined as a Matching Score) similarly associated with improved outcome. As the authors make clear in the discussion, this is not a randomized controlled study and could be influenced by potential selection bias; hence making it difficult to fully conclude that the benefit observed was specifically due to the MTB-recommended regimen administered. Although this needs to be taken into account, these are inherent challenges in assessing real-world clinical datasets of limited size.

The authors note that the results in this manuscript are consistent with prior analyses from the I-PREDICT trial where 73 patients received similar MTB-recommended regimens. However, there was one potentially notable difference in objective response (which is likely the best indicator whether the MTB-recommended regimen was active). In I-PREDICT, 45% of patients w/ \geq 50% matching scores achieved PR/CR, whereas 21% of patients achieved PR/CR in this manuscript. (the overall clinical benefit rate does not differ greatly due to the fact that only 5% patients had SD \geq 6 months in I-PREDICT, versus 31% in this report). Given that there appeared to be a broader range of biomarkers (notably IHC and mRNA highlighted Supplemental Tables 4 & 5) that contributed to treatment recommendations, it would be useful for the authors to comment on how the types of biomarkers may have influenced the outcome. (or were there other factors that influenced the lower PR/CR rate in this report compared to the initial I-PREDICT data?) For instance, was PR/CR or clinical benefit rate in the \geq 50% matching score group driven primarily by validated, genomically-defined targeted agents? How often were the less validated IHC tests used to guide a treatment recommendation and did they provide any value with respect to outcome?

Answer:

We thank reviewer for this insightful questions. As pointed out by the reviewer, although there were differences in response rate when compared to I-PREDICT study, overall clinical benefit rate was extremely similar—52% in the current study and 51% in the I-PREDICT study. The I-PREDICT study only used Foundation Medicine (now an FDA-approved companion diagnostic) for molecular profiling while, in this study, we have allowed any CLIA-certified (clinical-grade) lab, which may have affected the outcome. Indeed in this study, 42 patients received therapies that were partially matched to IHC testing (N=29 with Matching Score <50% and N=13 with Matching Score \geq 50). Because of the small number of patients, we could not determine if there as an effect on outcome. We believe this is part of the limitation and further validation is needed with a future study.

We have now commented following in the Discussion section:

Page 14:

“Lastly, differences in molecular profiling platforms may have affected the detection of targetable markers.”

Pages 14-15:

“Further clinical investigation is warranted in order to validate these findings as well as to determine if there are Matching Score thresholds that determine the utility of precision therapies.”

Compared to other reported MTB-guided precision medicine efforts, this study appears to represent one of the larger based upon the # of patients evaluable for therapy after MTB decisions. As such, the authors should comment on their learnings regarding the value of specific biomarkers. Nevertheless, the authors should at least provide some granularity around treatment recommendations, as this was not noted in the paper. A table similar to Supplementary Table 2 in their Nature Medicine 2019 paper would be informative to the reader (including Matching Score).

Answer:

We agree with the reviewer detailed clinical information.

We have now included Supplemental Data that includes patient demographics, cancer diagnosis, molecular information, treatment regimen, rate of matching, progression-free survival, overall survival and clinical benefit for the 429 patients evaluable for outcome.

We agree that this is one of the most comprehensive/largest molecular tumor board experienced. In addition to the supplemental data provided as noted above, one of the main points that we learned from this dataset is that the degree of matching was important.

Given that a majority of the patients who received physician’s choice regimens had matching scores <50% (ie. 95% of the 164 patients), was this associated with an earlier review during the 2012-2018 timeframe? (Put another way, the degree of biomarker content and more importantly the degree of confidence in the interpretation of biomarker/outcome associations would have been limited in the earlier years of this study, potentially leading to physician’s not following MTB recommendations)

Answer:

We thank the reviewer for this astute observation.

We have found that between December 2012-December 2015 vs. January 2016-September 2018, there was numeric increase in frequency of patients receiving therapy with high Matching Score (frequency of Matching Score \geq 50%: December 2012-December 2015, 15.3% [23/150] vs. January 2016-September 2018, 36.6% [102/279]). We agree with reviewer this is likely due to the changes in understanding and interpretation of biomarkers and outcomes along with the time.

We have now included this limitation at Discussion section, Pages 13-14 as following:

“(iv) the field of molecular oncology is rapidly evolving, hence the understanding and interpretation of biomarkers and outcomes can change. Indeed in this study, we have observed an increase in Matching Score over the time (frequency of Matching Score \geq 50%: December 2012-December 2015, 15.3% [23/150] vs. January 2016-September 2018, 36.6% [102/279]).”

What is the criteria for IHC/mRNA positivity? Given that it was utilized for treatment recommendations, it should be indicated.

Answer:

We thank the reviewer for this suggestion. IHC and RNA analyses were performed in several different laboratories but all were CLIA-certified and hence met the criteria for clinical grade testing in the USA. We have included the website in Supplemental Table 1 for the reference for the laboratories.

Since most IHCs were done at the Caris laboratory, we have also provided their criteria for IHC positivity in Supplemental Table 4. For specific markers such as ER, AR and HER2, positivity was consistent with NCCN guidelines. For the RNA expression analysis, it was mostly performed at Paradigm Diagnostics where the high mRNA expression was defined as > 5 fold change in expression as compared to a set of normal tissue controls (at least 6 normal tissue control from \sim 30 different primary tumor types). This information is now updated in Supplemental Table 5.

The authors should incorporate the number of centers patients treated who contributed to this study?

Answer: The study was done at the UCSD Moores Cancer Center and we have now included this in the Patients and Methods.

The prevalence of specific genomic aberrations was higher in the cfDNA dataset compared to the tissue NGS dataset. Of note, amplifications of BRAF, EGFR, CDK6, MET (as examples) approached ~7% in cfDNA, but were not identified in tissue DNA. There could be several explanations, including timepoint of sample acquisition in relationship to clinical treatment or technical given the differences in platforms. Since there are potentially actionable therapies for these specific examples, it would be informative to understand the criteria for determining amplification of genes on the panel if they were used for treatment recommendations.

Answer:

For Guardant cfDNA analysis, degree of amplifications were reported as follows: 1+, x2.13-2.40, which is the 10th to 50th percentile; 2+, x2.41-4.00, which is >50th to 90th percentile; and 3+, greater than x4.0 copy numbers, which is >90th percentile. We have included online information for cfDNA analysis in Supplemental Table 1.

We agree that the differences in the time point of molecular profiling as well as various molecular profiling platforms may have affected the outcome.

We have now commented following limitations in the Discussion section (Page 14):

Third, molecular analysis was obtained at various time points in relationship to the clinical history. Lastly, differences in molecular profiling platforms may have affected the detection of targetable markers.

Reviewers' Comments:

Reviewer #1:

Remarks to the Author:

Dear Authors,

I have reviewed your revised manuscript „Real-World Data from a Molecular Tumor Board: Improved Outcomes with a Precision N-of-One Strategy”.

I believe that it has considerably improved. There are only two comments remaining:

In the discussion section you state that

"... there were 10 patients who had a single gene alteration that was matched with targeted therapies and 8/10 (80%) patients had clinical benefit including N=5 partial responses (PFS: 8, 11+, 19, 20+ and 36 months) and N=3 durable stable disease \geq 6 months (PFS: 11+, 15 and 22+ months) (with "+" indicating ongoing responses). Further investigations with larger sample size are required to determine if certain combination approaches are more efficacious than others." - does the last sentence refer to the 10 patients with single-gene alterations only? Why are additional studies with combination therapy more relevant in this patient population than in the rest?

Could you briefly state if there are differences in the calculation of the matching score between the I-PREDICT and the current study? It is difficult to ascertain if e.g. RNA/IHC results (that were unfrequent in I-PREDICT except for hormone receptor) and multiple reports for the same patient were handled differently between the studies.

Reviewer #3:

Remarks to the Author:

I thank the authors for their responses to my questions and their diligence in compiling the supplemental data file. The responses were more than satisfactory. I find the paper acceptable for publication.

Reviewer #1 (Remarks to the Author):

Dear Authors,

I have reviewed your revised manuscript „Real-World Data from a Molecular Tumor Board:

Improved Outcomes with a Precision N-of-One Strategy”.

I believe that it has considerably improved. There are only two comments remaining:

Answer:

We thank the reviewer for the favorable comment.

1. In the discussion section you state that

“...there were 10 patients who had a single gene alteration that was matched with targeted therapies and 8/10 (80%) patients had clinical benefit including N=5 partial responses (PFS: 8, 11+, 19, 20+ and 36 months) and N=3 durable stable disease \geq 6 months (PFS: 11+, 15 and 22+ months) (with “+” indicating ongoing responses). Further investigations with larger sample size are required to determine if certain combination approaches are more efficacious than others.” - does the last sentence refer to the 10 patients with single-gene alterations only? Why are additional studies with combination therapy more relevant in this patient population than in the rest?

Answer:

We thank the reviewer for pointing this out. As noted, the order of the paragraph is awkward and we have corrected it as per below.

“These results are consistent with previous preliminary work from our group²⁷ as well as from our prospective trial--Investigation of Profile-Related Evidence Determining Individualized Cancer Therapy (I-PREDICT) [NCT02534675] trial, where 73 patients with treatment-refractory solid tumors were treated with a combination-based strategy using their unique tumor genomic signatures.¹⁵ Patients whose genomic aberrations were targeted with high Matching Scores demonstrated significantly better clinical outcomes including response rate, PFS, OS when compared to the lower matching group. Similar results were seen in the WINTHER trial (NCT01856296) where patients were navigated to therapy on the basis of DNA as well as RNA profiling.¹⁶ Further investigations with larger sample size are required to determine if certain combination approaches are more efficacious than others. Of note, in the current study, there were 10 patients who had a single gene alterations that were matched with targeted therapies, and 8/10 such patients (80%) attained clinical benefit including N=5 partial responses (PFS: 8, 11+, 19, 20+ and 36 months) and N=3 durable stable disease \geq 6 months (PFS: 11+, 15 and 22+ months) (with “+” indicating ongoing responses). Hence,

single gene targeting may be sufficient to achieve clinical response among patients harboring a limited number of genomic alterations.”

2. Could you briefly state if there are differences in the calculation of the matching score between the I-PREDICT and the current study? It is difficult to ascertain if e.g. RNA/IHC results (that were unfrequent in I-PREDICT except for hormone receptor) and multiple reports for the same patient were handled differently between the studies.

Answer:

The rules were handled in the same way in this study and in the I-PREDICT study, with a few caveats. First, rules for scoring were updated to account for emerging information in the literature. However, the specific updates did not affect any of the current patients because they did not have the specific alterations in the updates. Second, the rules for IHC and RNA were handled the same way in both studies but, as the reviewer notes, the RNA assays were available only in this study. (The IPREDICT study included only Foundation Medicine reports and IHC for hormone receptors). We state in the current paper: *“However, protein or RNA were only included in the calculation when they were targeted.”* Finally, in I-PREDICT, we used all data from the Foundation Medicine studies; in this paper, we did the following: *“If a patient had two or more tests from the same testing laboratory, the sample closest to the time of MTB was used for the analysis.”* The latter comment can be found in the footnote to Supplemental Table 1.

Reviewer #3 (Remarks to the Author):

I thank the authors for their responses to my questions and their diligence in compiling the supplemental data file. The responses were more than satisfactory. I find the paper acceptable for publication.

Answer: We thank the reviewer for the favorable comment.